# Neutrino-Mass Sensitivity and Nuclear Matrix Element for Neutrinoless Double Beta Decay

**Hiroyasu Ejiri** 

Research Center for Nuclear Physics, Osaka University, Osaka 567-0047, Japan; ejiri@rcnp.osaka-u.ac.jp

**Abstract:** Neutrinoless double beta decay (DBD) is a useful probe to study neutrino properties such as the Majorana nature, the absolute neutrino mass, the CP phase and the others, which are beyond the standard model. The nuclear matrix element (NME) for DBD is crucial to extract the neutrino properties from the experimental transition rate. The neutrino-mass sensitivity, i.e., the minimum neutrino-mass to be measured by the DBD experiment, is very sensitive to the DBD NME. Actually, the NME is one of the key elements for designing the DBD experiment. Theoretical evaluation for the DBD NME, however, is very hard. Recently experimental studies of charge-exchange nuclear and leptonic reactions have shown to be used to get single-$\beta$ NMEs associated with the DBD NME. Critical discussions are made on the neutrino-mass sensitivity and the NME for the DBD neutrino-mass study and on the experimental studies of the single-$\beta$ NMEs and nuclear structures associated with DBD NMEs.

**Keywords:** neutrinoless double beta decay; Majorana nature of neutrino; neutrino mass; nuclear matrix element; charge exchange reaction; ordinary muon capture; quenching of $g_A$

## 1. Introduction

Neutrinoless double beta decay (DBD), which violates the lepton-number ($L$) conservation law by $\Delta L = 2$, is beyond the standard electro-weak model. DBD is very powerful for studying neutrino properties such as the neutrino nature (Dirac or Majorana), the neutrino mass-scale and the mass-hierarchy, the neutrino phase, the possible right-handed weak interaction and the others, which are beyond the electro-weak standard model. Experimental and theoretical studies of DBDs have extensively been made for decades as given in review articles [1–5] and references therein.

The DBD transition rate is very sensitive to the DBD neutrino response (square of the DBD nuclear matrix element NME). Hereafter DBD and NME are used for the neutrinoless DBD and the neutrinoless DBD NME unless specified as two-neutrino DBD and two-neutrino DBD NME. The accurate value for the DBD NME is crucial for extracting the neutrino properties of particle physics interest from the DBD rate, if it is observed. The DBD neutrino-mass sensitivity, i.e., the minimum neutrino mass to be measured, depends on the NME. Thus one needs the NME as well to design the DBD experiment/detector. The DBD nuclear responses are discussed in review articles [6–8] and references therein, and the DBD NMEs are in reviews [9–11].

The neutrinoless DBD transition involves several transition modes such as the light neutrino-mass mode, the heavy neutrino-mass mode and others beyond the standard model [1,4]. The transition rate $T^{0\nu}$ for the $\alpha$-mode transition is expressed by a product of the nuclear physics factor $K_N(\alpha)$ and the neutrino physics factor $f_\nu(\alpha)$ as

$$T^{0\nu} = K_N \times |f_\nu(\alpha)|^2, \quad K_N = g_A^4 G^{0\nu} |M^{0\nu}|^2, \tag{1}$$

where $g_A$ = 1.27 is the axial-vector weak coupling for a free nucleon in units of the vector coupling $g_V$, $G^{0\nu}$ is the phase space volume and $M^{0\nu}$ is the neutrinoless DBD NME.

The phase space factor $G^{0\nu}$ is a kinematic factor, which depends on the $\beta\beta$ energy $E$ and the atomic number $Z$. It is proportional approximately to $E^5$ and increases as $Z$ increases. It also depends a little on the mode $\alpha$. The phase space has been well calculated [12–15].

The NME $M^{0\nu}$ reflects the internal nuclear structure, and depends on the mode $\alpha$ of the DBD process. In case of the light neutrino-mass mode, the transition rate depends on the neutrino effective mass, and the effective mass depends on the absolute neutrino mass, the mass hierarchy and the neutrino CP phases. Thus one needs the accurate value for the NME in order to study these neutrino properties.

The DBD event-rate $T^{0\nu}$ is extremely rare and the signal energy $E$ is very low. In the case of the DBD caused by the light Majorana neutrino with the effective mass $m^{eff} \approx 10$ meV, the rate is of the order of $10^{-29}$ per year (y). Then one needs the DBD isotopes of the order of $10^{29}$, i.e., $N \approx 10$ tons of the DBD isotope mass to get a few signals for a 5 y exposure. The DBD signal energy $E$ is around a few MeV, and thus is the same as the energy of typical background $\beta - \gamma$ rays. Therefore, one needs a very high-sensitivity low-background detector with the DBD isotope mass of the order of $N = 10$ tons.

Recent works on the neutrino oscillations show that the neutrinos are massive particles [16–18]. Then the neutrinoless DBD with the light neutrino-mass mode is very interesting to study the neutrino mass and the hierarchy. Actually, the DBD rate is proportional to the square of the effective neutrino mass $m^{eff}$, which depends on the CP phases and the mass hierarchy. So we discuss in the present report mainly the neutrino-mass mode DBD.

The DBD neutrino-mass sensitivity, which is defined as the minimum neutrino mass to be measured by the DBD experiment, is inversely proportional to the NME $M^{0\nu}$. The DBD NME is very sensitive to all kinds of nucleonic and non-nucleonic correlations, nuclear medium effects and effective weak-couplings. Accordingly, accurate evaluations for the NME by using existing theoretical nuclear models are indeed very hard. The DBD neutrino-mass sensitivity is discussed in the review [8] and also in a recent article [19].

Accordingly, experimental studies of the DBD NME have been made by using various kinds of nuclear and leptonic reactions. Recently charge-exchange nuclear reactions and muon capture reactions have been shown to be very useful to study the single-$\beta$ NMEs and the nuclear structures associated with the DBD NME. They provide useful nuclear parameters to be used for evaluating the DBD NME.

The present report aims to discuss briefly the neutrino-mass sensitivity and the NME for the DBD experiment to access the neutrino masses of the current interest and the experimental ways to study single-$\beta$ NMEs and nuclear structures associated with the DBD NME to help the theoretical evaluations for the DBD NME.

In Section 2, we discuss the DBD transition mode, the DBD NME for the neutrino-mass mode, and the DB neutrino-mass sensitivity. Experimental ways to study the single-$\beta$ NMEs associated with the DBD NME and nuclear structures relevant to the DBD are discussed in Section 3, and the quenching of the axial-vector weak coupling is discussed in Section 4. Finally, concluding remarks and discussions are given in Section 5.

## 2. DBD Transition Mode, DBD Neutrino-Mass Sensitivity and DBD NME

### 2.1. Neutrinoless DBD and Effective Neutrino Mass

The neutrinoless DBD, which requires the Majorana nature (neutrino = anti-neutrino) of the neutrino, is due to several modes ($\alpha$), the light neutrino-mass mode, the heavy neutrino-mass mode, the SYSY particle mode, the right-handed weak interaction mode, the Majoron-boson mode, and others. They are discussed as given in reviews [20–22] and references therein, and also in recent reviews [1,3–5] and references therein. The neutrinoless DBD is indeed a sensitive probe to study these neutrino

properties and weak interactions beyond the standard model. The neutrinoless DBD modes beyond the standard model, together the two neutrino DBD within the standard model, are schematically shown in Figure 1.

Double beta decay schemes

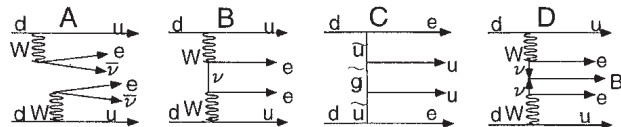

**Figure 1.** Schematic diagrams for 2 neutrino double beta decay (DBD) (**A**) and for neutrinoless DBDs with a Majorana neutrino exchange (**B**), the SUSY particle exchange (**C**) and the Majoron emission (**D**). DBD processes with 2 neutrons → 2 protons in the DBD nucleus are shown. d and u are d and u quarks in the 2 neutrons and the 2 protons, respectively [1,3].

In the case of the light neutrino-mass mode, a light Majorana neutrino is exchanged between the 2 neutrons in the DBD nucleus. The transition rate $T^{0\nu}$ is expressed by using the effective neutrino mass $m^{eff}$ [1,4,8] as

$$T^{0\nu} = g_A^4 G^{0\nu} |M^{0\nu}|^2 |m^{eff}|^2, \tag{2}$$

where $T^{0\nu}$ is written by using the halflife $t_{1/2}$ as $T^{0\nu} = \ln 2 / t_{1/2}$, $G^{0\nu}$ is the phase space volume for the light mass mode, $M^{0\nu}$ is the NME for the light neutrino-mass mode, and $m^{eff}$ is the effective neutrino mass, respectively. The effective neutrino mass is expressed as

$$m^{eff} = |\sum_i |U_{ei}|^2 m_i e^{i\alpha_i}| \tag{3}$$

where $U_{ei}$ is the mixing coefficient, $m_i$ is the $i$th eigen-mass and $\alpha_i$ is the phase, respectively. In cases of the normal mass-hierarchy (NH) of $m_1 \leq m_2 \ll m_3$ and the inverted mass-hierarchy (IH) of $m_3 \ll m_1 \leq m_2$, the effective masses are given approximately as

$$|m^{eff}| \approx |C_{13}^2 S_{12}^2 \sqrt{(\Delta m_s^2)} + S_{13}^2 \sqrt{(\Delta m_a^2)} e^{-2i\alpha_2}| \qquad \text{NH}, \tag{4}$$

$$|m^{eff}| \approx |C_{13}^2 \sqrt{(\Delta m_a^2)} [1 - \sin^2 2\theta_{12} \sin^2 \alpha_{12}]^{1/2}| \qquad \text{IH}, \tag{5}$$

where $C_{ij}$ and $S_{ij}$ are the mixing coefficients, $\theta_{ij}$ is the mixing angle, $\alpha_{12} = \alpha_2 - \alpha_1$ is the Majorana phase difference, $\Delta m_s^2$ is the solar neutrino mass-square difference, and $\Delta m_a^2$ is the atmospheric neutrino mass-square difference [1,4,5,7,8].

The mixing parameters, the mixing angles and the mass-square differences in Equations (4) and (5) are known from the neutrino oscillation experiments. Then the effective mass to be studied is given as a function of the minimum neutrino mass $m_0 = m_1$ and the Majorana phase $\alpha_2$ in the case of NH and that of the minimum mass $m_0 = m_3$ and the Majorana phase difference $\alpha_2 - \alpha_1$ in the case of IH, respectively. The regions of the effective neutrino-masses are shown as a function of $m_0$ in Figure 2. Here the constrains from the cosmological bound of $m_0 \leq 9$ meV for IH and $m_0 \leq 26$ meV for NH are also shown in Figure 2.

The upper and lower bounds correspond to the effective masses for $\alpha_2 = 0$ and $\alpha_2 = \pi/2$ in the case of NH, and for $\alpha_{12} = 0$ and $\alpha_{12} = \pi/2$ in the case of IH. Thus the experimental value for the effective mass depends on the mass hierarchy, the minimum mass and the Majorana phases.

Then one gets $m^{eff} \approx 50 - 18$ meV, depending on $\alpha_{12} = 0 - \pi/2$, and $m_0 \leq 9$ meV in the case of the IH, and $m^{eff} \leq 15$ meV and $m_0 \leq 26$ meV in the case of the NH. The effective mass in the case of IH is plotted as a function of $\alpha_{12}$ in Figure 3. If the measured effective mass is larger than 18 meV, one concludes the IH mass spectrum and gets the Majorana phase $\alpha_{12}$. If that is smaller than 4 meV,

one concludes the NH mass spectrum and gets constrains on the minimum mass $m_0$ and the Majorana phase of $\alpha_2$.

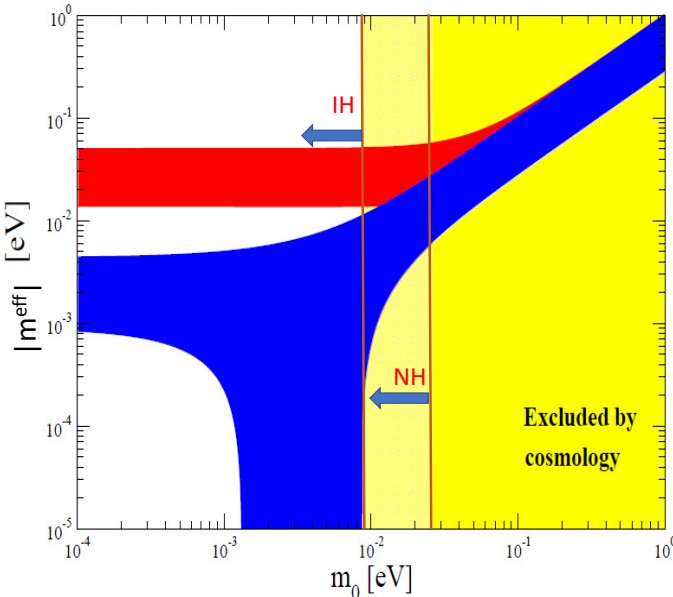

**Figure 2.** Effective neutrino masses $m^{eff}$ in the cases of normal mass-hierarchy (NH) and inverted mass-hierarchy (IH) as a function of the minimum mass $m_0$ and the mass constrains from the cosmological bound of $\sum m_i \leq 110$ meV.

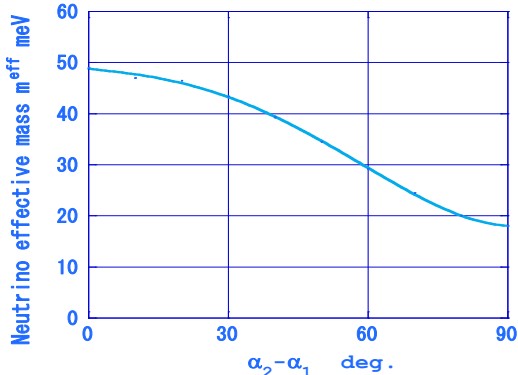

**Figure 3.** Effective neutrino masses $m^{eff}$ in the case of IH as a function of the phase difference $\alpha_2 - \alpha_1$.

*2.2. DBD Neutrino-Mass Sensitivity and DBD Nuclear Matrix Element*

The DBD transition rate is rewritten for practical use in terms of the number of the DBD signals per ton (t) of the DBD isotopes per year (y). It is expressed by using the nuclear sensitivity $m_n$ as [1,8]

$$T^{0\nu} = \left(\frac{m^{eff}}{m_n}\right)^2, \quad m_n = \frac{k}{M^{0\nu}}, \quad k = 7.8\text{meV}\frac{A^{1/2}}{g_A^2 (G^{0\nu})^{1/2}}, \tag{6}$$

where $T^{0\nu}$ is the transition rate in units of per t y, $k$ is the nuclear kinematic factor defined as the nuclear sensitivity $m_n$ in the case of $M^{0\nu} = 1$, and $G^{0\nu}$ is the phase space volume in units of $10^{-14}/\text{y}$, and $A$ is the mass number of the DBD nucleus. $m_n$ corresponds to the effective neutrino mass to give the DBD rate of $1/\text{t}$ y. The factor $k$ reflects the external kinematic factor and $1/M^{0\nu}$ does the internal nuclear-structure one, and the nuclear sensitivity $m_n$ is given by the product of them.

The nuclear kinematic factor $k$ and the nuclear sensitivity $m_n$ for typical DBD nuclei of current interests are shown in Table 1. It is interesting to note that the kinematic factors $k$ for most nuclei

are around 40 meV near the upper bound of the IH mass, while the values for $^{76}$Ge and $^{150}$Nd are, respectively, a factor of two larger and smaller than 40 meV because of the smaller and larger phase space factors. The $m_n$ = 40 meV sensitivity means one DBD signal per year per ton of the DBD isotopes in the case of $M^{0\nu} = 1$ and $m^{eff}$ = 40 meV, and four signals in the case of $M^{0\nu} = 2$.

**Table 1.** Nuclear sensitivities for DBD nuclei of the current interest. Shown are the isotope, its natural abundance (N.a.), the $Q$ value, and the nuclear sensitivity. $k$: the nuclear kinetic factor, $m_n^*$: nuclear sensitivity in the case of $M^{0\nu} = 2$, and $m_n^{**}$: nuclear sensitivity in the case of $M^{0\nu} = 3$.

| Nuclide | N.a. (%) | Q Value (keV) | k meV | $m_n^*$ meV | $m_n^{**}$ meV |
|---|---|---|---|---|---|
| $^{76}$Ge | 7.44 | 2039 | 80.8 | 40.4 | 26.7 |
| $^{82}$Se | 8.73 | 2997 | 40.0 | 20.0 | 13.3 |
| $^{100}$Mo | 9.63 | 3034 | 34.8 | 17.4 | 11.6 |
| $^{116}$Cd | 7.49 | 2814 | 36.0 | 18 | 12 |
| $^{130}$Te | 33.8 | 2528 | 40.2 | 20.1 | 13.4 |
| $^{136}$Xe | 8.9 | 2458 | 40.2 | 20.1 | 13.4 |
| $^{150}$Nd | 5.64 | 3371 | 19.8 | 9.9 | 6.6 |

The DBD neutrino-mass sensitivity $m_m$ is derived from the condition that the number of the DBD signals is well above the fluctuation of the number of background (BG) signals. Then one gets

$$T^{0\nu} NT\eta\epsilon \geq \delta_0 (BNT)^{1/2}, \tag{7}$$

where $N$ is the DBD isotope mass in units of ton (t), $\eta$ is the enrichment, $\epsilon$, is the signal efficiency, $T$ is the exposure time in units of year (y), $B$ is the BG rate per t y, and $\delta_0$ is a constant around two in the case of the 90% confidence level for the DBD signal identification. The DBD neutrino-mass sensitivity is given as [1,8]

$$m_m = m_n d, \qquad d = d_0 \times \eta^{-1/2} \epsilon^{-1/2} (NT/B)^{-1/4}, \tag{8}$$

where $d$ is the detector sensitivity and $d_0$ is around $\sqrt{\delta_0} \approx 1.4$.

The DBD neutrino-mass sensitivity depends on the nuclear sensitivity $m_n = k/M^{0\nu}$, and thus linearly on $M^{0\nu}$, but weakly on $N$ and $B$ by the power of 1/4. Thus the 30% less $M^{0\nu}$ is equivalent to an order of magnitude more isotope mass $N$ or less BG rate $B$. Thus one needs an accurate value for the NME $M^{0\nu}$ to design the DBD detector with the given neutrino-mass sensitivity.

The DBD NME is expressed as [7,8]

$$M^{0\nu} = (\frac{g_A^{eff}}{g_A})^2 [M_{GT}^{0\nu} + M_T^{0\nu}] + (\frac{g_V^{eff}}{g_A})^2 M_F^{0\nu}, \tag{9}$$

where $M_{GT}^{0\nu}$, $M_T^{0\nu}$ and $M_F^{0\nu}$ are the Gamow–Teller ($GT$), Fermi ($F$) and tensor ($T$) NMEs, respectively, and $g_A^{eff}$ and $g_V^{eff}$ are the effective axial-vector and vector couplings in units of the vector coupling $g_V$ for a free nucleon. The GT, T and F NMEs are given as

$$M_{GT}^{0\nu} = \sum_i < t_\pm \sigma h_{GT}(r_{12}, E_i) \sigma t_\pm >, \tag{10}$$

$$M_T^{0\nu} = \sum_i < t_\pm h_T(r_{12}, E_i) S_{12} t_\pm >, \tag{11}$$

$$M_F^{0\nu} = \sum_i < t_\pm h_F(r_{12}, E_i) t_\pm >, \tag{12}$$

where the sum is for all the intermediate states (i), $t$ and $\sigma$ are the isospin and spin operators, and $h_\alpha$ is the neutrino potential with $\alpha = GT, T, F$, and $r_{12}$ is the distance between the two neutrons involved in the virtual-neutrino exchange.

The NMEs of $M_{GT}^{0\nu}$, $M_T^{0\nu}$ and $M_F^{0\nu}$ are evaluated as given in Equations (10)–(12) by using appropriate theoretical models. Then the $g_A^{eff}$ and $g_V^{eff}$ stand for the quenching (re-normalization) effects due to such nucleonic and non-nucleonic correlations and nuclear medium effects that are not explicitly included in the model.

The neutrino potential is approximately given in terms of $1/r_{12} \approx p$ with $p$ being the virtual-neutrino momentum of the order of 50–150 MeV/c. Then the angular momentum is around $l\hbar = (1\text{--}4)\,\hbar$. Accordingly, the GT and T NMEs are very sensitive to nuclear spin isospin and momentum correlations and the F NME to the isospin and momentum correlations. In the case of a typical DBD run for $T = 5$ y by using a typical detector with $N = 1$ t, $B = 1/$t y, $\eta = 0.9$ and $\epsilon = 0.6$, one gets the detector sensitivity $d \approx 1.5$ and the neutrino mass sensitivities $m_m \approx 1.5\, m_n$.

The DBD neutrino-mass sensitivities $m_m$ in the cases of the NMEs of $M^{0\nu} = 1, 2$ and the BG rates of $B = 1/$t y and 0.01 /t y are shown as a function of the exposure of $NT$ in Figure 4. Then, one needs exposures of $NT \approx 3, 16$ and 256 t y in order to access the effective mass of $m_m = 20$ meV in cases of the NMEs of $M^{0\nu} = 3, 2$ and 1, respectively. One needs almost a 1000 ton year exposure to access the NM mass in the case of $M^{0\nu} = 2$ and $B = 0.01/$t y. The NME is indeed a key parameter for the neutrino mass study by DBD experiments.

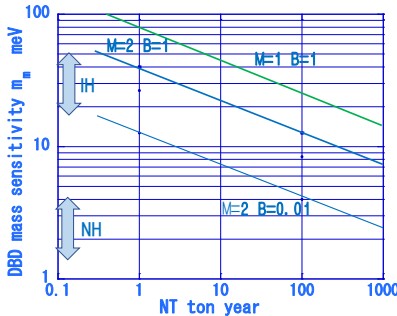

**Figure 4.** Double beta decay (DBD) neutrino-mass sensitivities $m_m$ in the cases of the NMEs $M^{0\nu} = 1, 2$ and $B = 1/$t y and 0.01/t y as a function of the exposure $NT$. Here $k = 40$ meV, $\eta = 0.9$ and $\epsilon = 0.6$ are assumed. The arrows with IH and NH are the mass regions for the IH and NH cases.

Experimentally one gets the value for or the upper limit on $M^{0\nu} \times m^{eff}$ from the measured value for or the upper limit on the DBD rate, i.e., the value for or the lower limit on the halflife. The present limits on the DBD halflives are discussed in the recent conference report [23]. Quite stringent limits on the DBD rates, $\ln 2/t_{1/2}$, have been obtained for $^{76}$Ge [24,25], $^{130}$Te [26], $^{136}$Xe [27,28] and others. Figure 5 shows the allowed regions for $M^{0\nu} \times m^{eff}$ in the case of the half life limit of $T_{1/2} \leq 10^{26}$ y for $^{76}$Ge and $^{136}$Xe, which is close to the present limits [25,28]. Therefore, experimental groups are trying to improve their detector sensitivities to access the IH neutrino-mass by increasing the isotope mass $N$ and/or lowering the BG rate $B$. The improvement factor does depend much on the NME.

In fact, the DBD nucleus of the current interest is a medium-heavy nucleus with the mass number $A \approx 100$. It is a complex many-body hadron (nucleon, meson, and isobar) system, and thus the DBD NME is very sensitive to all kinds of nucleonic and non-nucleonic correlations, nuclear deformations and nuclear medium effects. Therefore, accurate theoretical evaluations for the DBD NME $M^{0\nu}$, including $g_A^{eff}$ and $g_V^{eff}$, are in practice very hard. They depend much on the models and the interaction parameters used for the models. Accordingly, the evaluated values for the DBD NMEs scatter over an order of magnitude, depending on the model, the $g_A^{eff}$ and other parameters used in the model [7,8]. Here it is noted that the current neutrino-mass limits are derived from the observed limits on the DBD halflives by using the existing minimum and maximum theoretical NMEs, which are sensitive to the models and parameters used in the calculations.

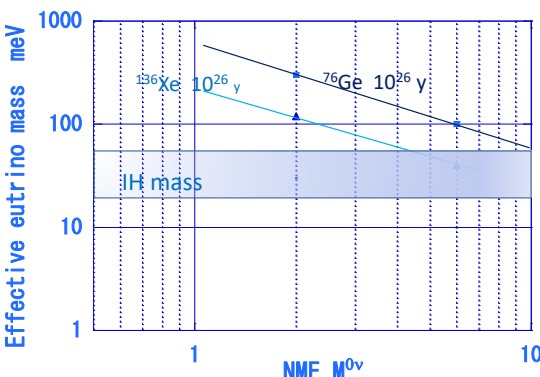

**Figure 5.** Effective neutrino mass $m^{eff}$ and NME $M^{0\nu}$. IH: the IH effective mass. Two lines show upper limits in the case of the halflife limits of $10^{26}$ y for $^{76}$Ge and $^{136}$Xe.

## 3. Experimental Approaches to DBD NMEs

DBD NMEs are very important for DBD neutrino studies, but accurate theoretical evaluations for them are very hard, and there are no direct ways to measure accurately the DBD NME $M^{0\nu}$. Recently various experimental studies have been made on single $\beta$-NMEs and nuclear structures associated with the DBD NME to provide useful information to help the theoretical evaluations for the DBD NMEs.

### 3.1. Experimental Studies of Two-Neutrino DBD NMEs

Here we note the 2-neutrino DBD ($2\nu\beta\beta$) NMEs for the transition of $^A_Z X \rightarrow ^A_{Z+2} X$, where $A$ is the mass number of the DBD nucleus and $Z$ and $Z+2$ are the atomic numbers of the initial and final nuclei, respectively. The transition rate $T^{2\nu}$ is expressed as

$$T^{2\nu} = g_A^4 G^{2\nu} |M^{2\nu}|^2, \tag{13}$$

where $G^{2\nu}$ is the phase space volume, $M^{2\nu}$ is the 2-neutrino NME. The NME is given by

$$M^{2\nu} = \sum_i \frac{M_{GT}^-(i) \times M_{GT}^+(i)}{\Delta(i)}, \tag{14}$$

where $M_{GT}^-(i)$ and $M_{GT}^+(i)$ are, respectively, the single-$\beta^-$ NME for $^A_Z X \rightarrow ^A_{Z+1} X$ and the single-$\beta^+$ NME for $^A_{Z+2} X \rightarrow ^A_{Z+1} X$ via the $i$th state in the intermediate nucleus, and $\Delta(i)$ is the energy denominator expressed as $\Delta(i) = Q_i + Q_{\beta\beta}/2$ with $Q_i$ and $Q_{\beta\beta}$ being the EC (e-capture) Q value for the $i$th intermediate state and the DBD Q value.

Actually, the 2-neutrino DBD NMEs have been extensively studied experimentally and theoretically, as given in recent reviews [8,11]. Some of the theoretical works are QRPA model calculations [29–31], shell-model calculations [32–35], IBM-model calculations [36], and effective field theory calculations [37]. Here we show the 2-neutrino DBD NMEs evaluated empirically by using experimental single-$\beta$ NMEs for low-lying (fermi-surface) quasi-particle (FSQP) states [38]. Then the single-$\beta^\pm$ NMEs are expressed as

$$M_{GT}^\pm(i) = k_Q^\pm M_{GT-QP}^\pm(i), \tag{15}$$

where $M_{GT-QP}^\pm(i)$ is the single-$\beta^\pm$ GT NME for the $i$th FSQP, and $k_Q^\pm = (g_{A-QP}^{eff}/g_A)^\pm$ stands for the quenching (re-normalization) coefficient for the $\beta^\pm$ due to the nucleonic and non-nucleonic correlations and nuclear medium effects, which are not included in the simple QP model. The quenching coefficient

$k_Q^\pm$ is derived as the ratio of the experimental $M_{GT}^\pm(i)$ and the FSQP model $M_{GT-QP}^\pm(i)$ by assuming the same phase for both $\beta^+$ and $\beta^-$ NMEs.

The NMEs $M^{2\nu}$ for DBD nuclei of the current interests are derived by using the FSQP model with the experimental $g_{A-QP}^{eff}/g_A$ and $\Delta(i)$ [4,8,38]. They reproduce well the NME $M^{2\nu}$ derived from the observed rate $T^{2\nu}$ as shown in Figure 6.

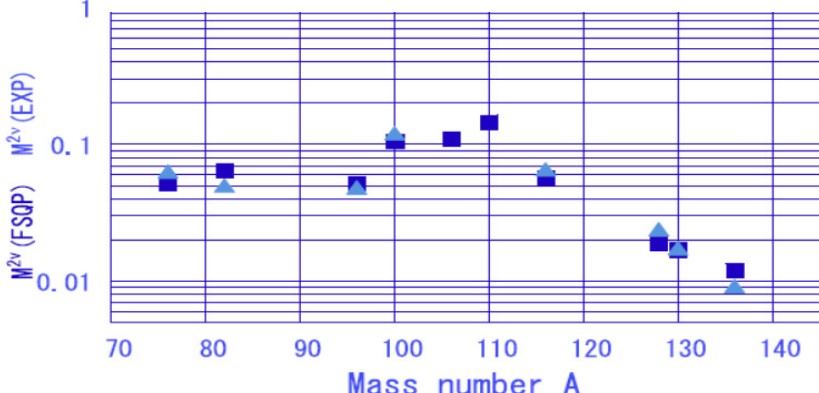

**Figure 6.** Two-neutrino DBD NMEs (triangles ) derived from the observed 2-neutrino DBD rates and the (fermi-surface) quasi-particle (FSQP) model NMEs (squares) based on the experimental quenching coefficients for single-$\beta$ GT NMEs [8,38].

On the other hand, the neutrinoless DBD NME $M^{0\nu}$ includes the neutrino potential $h_k(r_{12})$ to rink the single-$\beta^\pm$ NMEs, as expressed in Equations (10)–(12). Among the three DBD modes, the GT and F mode NMEs of $M_{GT}^{0\nu}$ and $M_F^{0\nu}$ are major components. The axial-vector single-$\beta^\pm$ NMEs and the vector single-$\beta^\pm$ NMEs are then associated, respectively, with the $M_{GT}^{0\nu}$ and $M_F^{0\nu}$ via the neutrino potential. Accordingly, the single-$\beta^\pm$ NMEs are used to help theoretical evaluations for $M_{GT}^{0\nu}$ and $M_F^{0\nu}$.

Actually, single-$\beta^\pm$ NNEs and nuclear structures associated with the DBD NMES are experimentally studied by means of nuclear charge-exchange reactions (CERs), leptonic (muon and neutrino) CERs and photo-nuclear reactions.

### 3.2. Nuclear Charge-Exchange Reactions

Nuclear CERs are used to study single-$\beta^\pm$ NMEs in wide energy and momentum regions relevant to DBD. The high energy-resolution ($^3$He,t) CER at RCNP Osaka is a powerful probe to study axial-vector NMEs for individual states in DBD nuclei. Axial-vector states are preferentially excited by using the 0.42 GeV $^3$He beam. GT NMEs with $J^\pi = 1^+$ have been measured, as shown in Figure 7, and used to evaluate the two-neutrino DBD NMEs [38–41].

The spin-dipole (SD) NME with $J^\pi = 2^-$ is one of the major components of the neutrinoless DBD NMEs. Recently the SD NMEs were derived for the first time from the CER cross sections [42,43]. They are found to be consistent with the FSQP model NMEs and the NMEs derived from the $\beta$-decay $ft$ values, as shown in Figure 8. Here it is noted that the momentum dependence of the NMEs was studied by measuring the cross sections in a wide angular range of $\theta = 0$–4 deg, corresponding to the momentum transfer of $p = 20$–120 MeV/c relevant to the DBD momentum region [44]. The observed GT and SD responses (squares of the GT and SD NMEs) are found to be constant over the wide momentum range, and thus one can use both the responses for the single-$\beta$ decays and the two-neutrino DBDs at the low momentum region and those for the CERs at the medium momentum region for evaluating the neutrinoless DBD responses [43,44].

The CER energy spectra show weak discrete peaks at the low energy region and the strong GT and SD giant resonances at the medium energy region, as shown in Figure 7. Note that the GT and SD strengths for low-lying states are much reduced with respect to the single quasi-particle strengths due to the destructive interference with the strong GT and SD GRs [6–8].

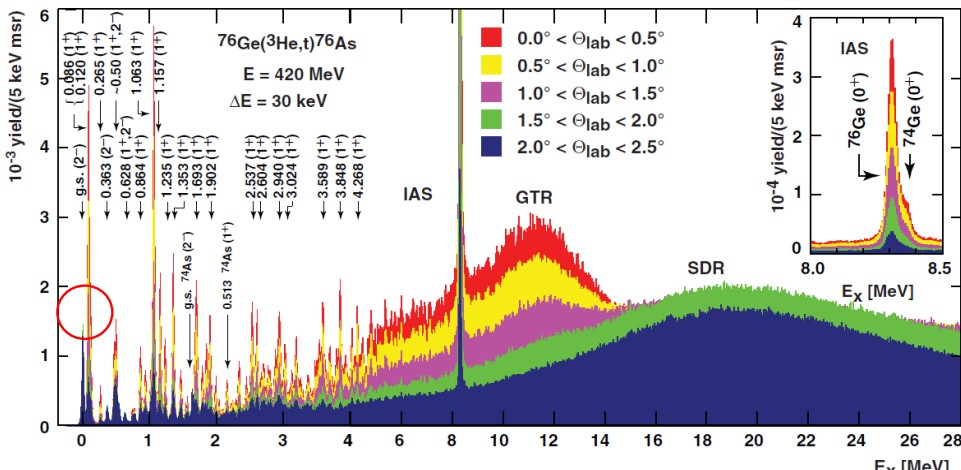

**Figure 7.** Energy spectrum of the ($^3$He,t) charge-exchange reactions (CER) on $^{76}$Ge. The spin-dipole (SD) (blue) and Gamow–Teller (GT) (red/yellow) peaks are well separated, as shown in the red circle [39].

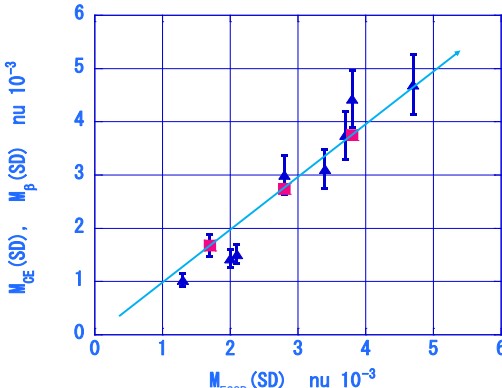

**Figure 8.** SD NMEs for DBD nuclei. The SD NMEs $M_{CE}$(SD) (blue triangle) measured by the ($^3$He,t) CERs and the NMEs $M_\beta$(SD) (red square) derived from the $\beta$ decay $ft$ values are plotted against the FSQP NMEs $M_{FSQP}$(SD).

### 3.3. Leptonic (Muon and Neutrino) Charge-Exchange Reactions

Low-momentum negative muons stopped in a target are trapped into the lowest orbit of the target atom. Then, they are captured into the nucleus via the leptonic CER of $\mu^- + {}^A_{Z+2}X \to \nu_\mu + {}^A_{Z+1}X^*$. This is analogous to the anti-neutrino nuclear interaction of $\bar{\nu}_\mu + {}^A_{Z+2}X \to \mu^+ + {}^A_{Z+1}X^*$. Muon capture reactions and nuclear structures are reviewed in [45]. The $\mu$ CER transfers the medium energy and medium momentum relevant to DBD. Thus the muon CER is used to study the NMEs relevant to DBD in the same energy and momentum regions. The muon CER is schematically shown in Figure 9.

The muon-capture reaction on $^{100}$Mo shows a giant resonance around $E \approx 12$ MeV [46], as shown in Figure 9. The observed GR is reproduced by the pnQRPA calculation [47]. The pnQRPA calculations for the capture rates for $^{100}$Mo and other medium-heavy nuclei are much smaller than the empirical rates [48], suggesting some quenching coefficient of $g_A^{eff}/g_A \approx 0.4$ [47,49], as in the $\nu$ NMEs studied by the nuclear $\beta^\pm$ decays [50,51]. On the other hand the QRPA calculations for medium heavy nuclei are in accordance with the empirical rates, suggesting no quenching for $g_A^{eff}$ [52]. Since the capture

rate is sensitive to the strength distribution as a function of the excitation energy, it is interesting to compare the theoretical and experimental distributions.

DBD NMEs are studied directly by using intense electron-neutrino beams obtained from high-intensity GeV-proton accelerators as the ORNL 1 GeV SNS and the J-PARC 3 GeV booster-synchrotron [53,54]. The neutrino energy spectrum is shown in Figure 10.

The neutrino energy of around 10–45 MeV is just appropriate for studying the NMEs for the medium energy neutrinos relevant to DBD. The expected neutrino intensity is of the order of $10^{15}$ per sec. Since the neutrino cross-section is of the order of $10^{-40}$ cm$^2$, one needs multi-ton scale DBD isotopes as used for DBD experiments.

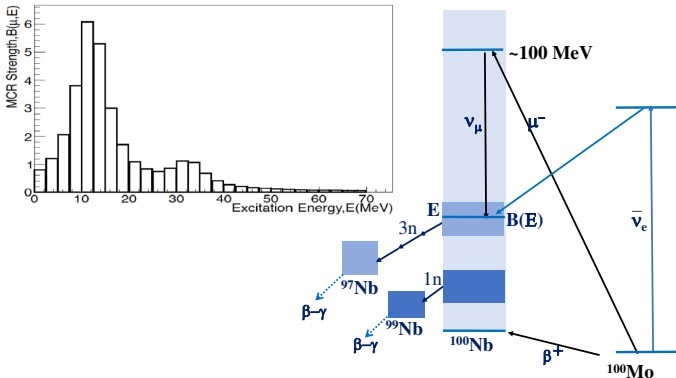

**Figure 9.** Schematic diagram of muon CER and the transition rate as a function of excitation energy.

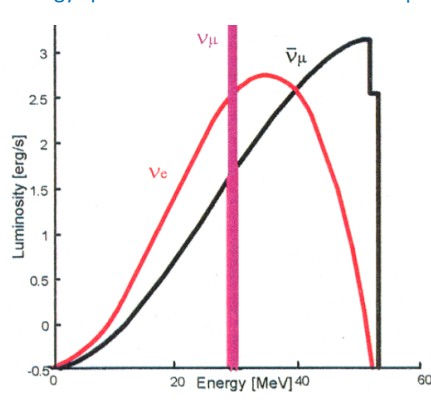

**Figure 10.** Energy spectrum for neutrinos from decays of $\pi^+$ and $\mu^+$ produced by the SNS GeV-proton nuclear interactions.

### 3.4. Photo-Nuclear Reactions

Electromagnetic (EM) interactions are used to study DBD NMEs. The charged-current weak NME is derived from the isovector component of the EM NME. The electric and magnetic EM NMEs correspond to the axial-vector and vector weak NMEs, respectively. The special case is the EM NME for the isobaric analog state (IAS), as shown in Figure 11. The EM and weak NMEs are related as [8,55,56]

$$< f|g_W m^\beta|i >= \frac{g_W}{g_{EM}}(2T)^{1/2} < f|g_{EM}m^\gamma|IAS >, \tag{16}$$

where $g_W$ and $g_{EM}$ are the weak and EM couplings, respectively, $T$ is the isospin of the initial state and $m^\beta$ is the weak transition operator analogous to the EM one $m^\gamma$. The $\beta$-NME analogous to the E1-$\gamma$ NME was obtained for $^{141}$Ce. The obtained NME is found to be quenched by a factor of 0.3 with respect to the QP NME [8,55,57].

NMEs for $|f\rangle \rightarrow |i\rangle$ are studied by measuring neutrons following photo-nuclear reactions of $(\gamma, n)$ on $|f\rangle$ via IAS of $|i\rangle$ as illustrated in Figure 11. The photo-nuclear reaction cross-section is used to get the absolute value for the NME, and the angular correlation of the $(\gamma, n)$ from the polarized photon is used to get the multi-polarity (E1,M1 or E2) of the NME [56]. Here the polarized photon is obtained from the polarized laser photon scattered off the GeV electron (Figure 12).

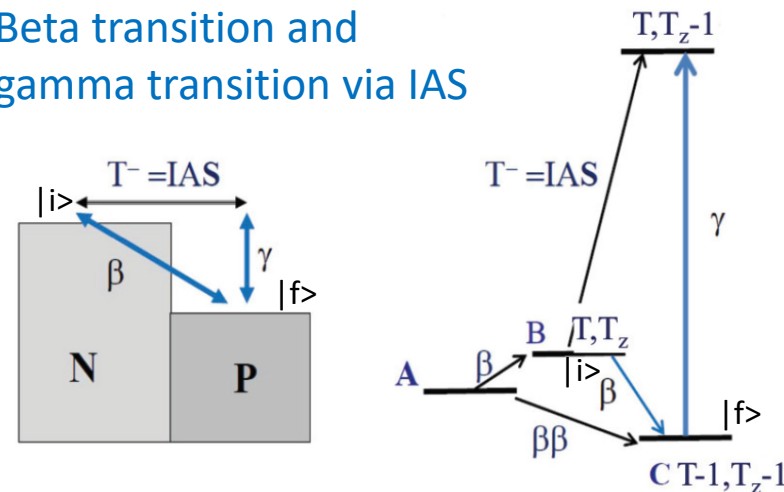

**Figure 11.** Schematic diagram of $\beta$ decay from $|i\rangle$ to $|f\rangle$ and $\gamma$ transition from $|f\rangle$ to the isobaric analog state (IAS) state $|IAS\rangle = T^- |i\rangle$ with $T^-$ being the isospin lowering operator. They are analogous with each other.

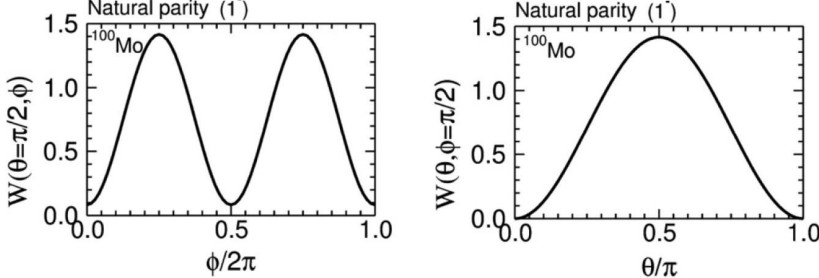

**Figure 12.** Angular distributions of photo-neutrons via the isobaric analog states from polarized photons on $^{100}$Mo [8,56].

## 4. Quenching Coefficient for Weak Coupling and DBD NME

Axial-vector weak transition rates for low-lying states are quenched (reduced) with respect to simple quasi-particle (QP) model evaluations, as discussed since the 1960s [6,7]. Similarly, magnetic gamma-transition rates and magnetic moments for low lying states are also quenched (reduced) with respect to QP model evaluations. These are due to the nucleonic and non-nucleonic spin-isospin correlations and others, as discussed in review articles [6,8,57–59].

The quenching coefficient is conventionally expressed in terms of the effective axial-vector coupling $g_A^{eff}/g_A$. It is derived from the experimental and theoretical single-$\beta$ NMEs as

$$\frac{g_A^{eff}}{g_A} = \frac{M_{\mathrm{EXP}}}{M_{\mathrm{MODEL}}}, \tag{17}$$

where $M_{EXP}$ and $M_{MODEL}$ are the experimental and theoretical (model) NMEs, respectively. Here we use the pnQRPA model NME $M_{QR}$, which includes the spin-isospin nucleonic correlations.

The quenching coefficients to be discussed are for medium-heavy nuclei of the DBD interest. The coefficients for axial vector GT and SD NMEs derived from the low-momentum (1–3 MeV/c) $\beta$-decay rates are around $g_A^{eff}/g_A \approx 0.5$ [50,51] as shown in Figure 13. The ($^3$He,t) CERs show that the GT and SD NMEs remain constant over the wide momentum region of $p$ = 30–100 MeV/c relevant to the DBD momentum [44], and the coefficients for GT and SD transitions are around 0.5. The muon CER covers the similar momentum region of $p \approx 80$ MeV/c. The muon capture rates on $^{100}$Mo and other DBD nuclei show the similar quenching coefficient around 0.4–0.5 [47,49], but the recent QRPA calculations suggest no quenching [52]. The magnetic hexadecapole (HD) NMEs, which are mainly isovector component, are also quenched by a coefficient around 0.33 with respect to the multi quasi-particle phonon (MQPP) model [60].

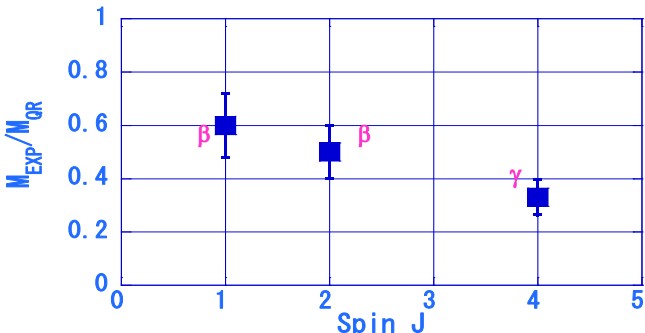

**Figure 13.** Quenching coefficients $g_A^{eff}/g_A$ and the spin *J* for GT(*J* = 1), SD (*J* = 2) and HD (*J* = 4) NMEs. $M_{EXP}$ and $M_{QR}$ are the experimental and pnQRPA model NMEs, respectively.

In the present work, we discussed mainly the quenching coefficient with respect to QRPA. The quenching of the axial-vector NMEs have been discussed in other model calculations as discussed in [8]. Among them, IBM-2 [61] gives similar quenching coefficients as in the present work. It is of great interest to study experimentally and theoretically how the quenching coefficients are universal or depend on the momentum and multi-polarity in medium heavy nuclei. It is noted that the quenching coefficient $g_A^{eff}/g_A$ reflects some nuclear core effects such as non-nucleonic (isobar) and nuclear medium effects, which manifest in medium and heavy nuclei.

The neutrinoless DBD NMEs for $^{130}$Te are plotted as a function of the quenching coefficient $g_A^{eff}/g_A$ in Figure 14. Here we use the GT, F, and F NMEs evaluated by pnQRPA [62] and the quenching coefficient $g_V^{eff}/g_V$ = 1 for the vector coupling. The GT and T NMEs decrease as the quenching coefficient decreases, while the F NME stays constant. In the case of the experimental coefficient of $g_A^{eff}/g_A$ = 0.5, one gets $M^{0\nu}$ = 1.8. Similarly, one gets $M^{0\nu}$ = 2.4 for $^{76}$Ge. These NMEs are based on the pnQRPA NMEs [62] and the experimental $g_A^{eff}$, and thus include uncertainties, which should be further investigated.

Here it is noted that 2-neutrino DBD NMEs (particularly $M^+$ in Equation (14)) are sensitive to the particle-particle coupling $g_{pp}$ because the low-momentum GT proton→neutron transitions are blocked by the neutron excess [51]. However, neutrinoless DBD NMEs involve mainly medium-momentum and medium-multipole transitions and thus the NMEs are not sensitive to $g_{pp}$.

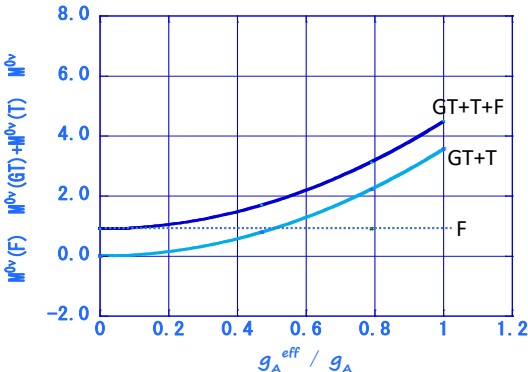

**Figure 14.** Neutrinoless DBD NMEs as a function of the quenching coefficient $g_A^{eff}/g_A$ for $^{130}$Te. $M^{0\nu}$: GT + T + F. $M^{0\nu}$(GT) + $M^{0\nu}$(T): GT + T. $M^{0\nu}$(F): F.

### 5. Concluding Remarks and Discussions

The neutrinoless DBD is a very sensitive and powerful probe for studying neutrino properties of particle physics interests beyond the standard model. Extensive experimental and theoretical studies of the DBDs are under progress to study the neutrino masses in IH and NH. The neutrino-mass sensitivity $m_m$ depends on the nuclear sensitivity $m_n$ and the detector sensitivity $d$, and the nuclear sensitivity depends on the phase space factor and the nuclear matrix element NME. Actually, the $m_m$ is very sensitive to the NME, which is one of the key elements of the high-sensitivity DBD experiments to search for the very small $\nu$ mass. There are, however, no direct experimental ways to measure the DBD NME, and accurate theoretical evaluations for the NME, including the weak coupling constant $g_A$, are very hard.

The high energy-resolutions nuclear CER of ($^3$He,t) and the leptonic CER of ($\mu$, $\nu_\mu$) at RCNP Osaka have recently been shown to be useful for experimental studies of the DBD NMEs. These reaction data provide the single-$\beta$ NMEs and the nuclear structures associated with the DBD NMEs.

The single-$\beta$ NMEs in the wide momentum region of the DBD interest are found to be quenched uniformly by the coefficient $g_A^{eff}/g_A \approx 0.5$ with respect to the pnQRPA NMEs due to such non-nucleonic correlations and nuclear medium effects that are not explicitly included in the pnQRPA. Using this quenching coefficient, the DBD NMEs are evaluated as $M^{0\nu} \approx 2.4$ and 1.8 for the DBD nuclei of $^{76}$Ge and $^{130}$Te, respectively. The halflives in the case of the IH mass of $m^{eff} = 20$ meV are $1.5 \times 10^{28}$ y and $0.4 \times 10^{28}$ y for $^{76}$Ge and $^{130}$Te, respectively.

The single-$\beta$ NMEs relevant to the DBD NMEs are studied also by neutrino nuclear reactions using neutrino beams and photo nuclear reactions using polarized photon beams as well. Nuclear structures and NMEs relevant to the DBD NMEs may be studied by double charge-exchange reactions using light and heavy ion beams [63] at INFN-LNS Catania, RCNP Osaka, RIKEN Wakoh and others as discussed in [8].

The quenching problem has been discussed mainly on axial-vector NMEs, and thus the quenching coefficient is discussed in terms of the effective (re-normalized) axial-vector weak coupling $g_A^{eff}$. In fact, medium-heavy DBD nuclei are a very complex many-body hadron (nucleon/mason/isobar) system, but most models are such simplified nucleon-based models that do not include non-nucleonic correlations, higher-shell orbits and/or nuclear medium effects. Accordingly, these effects are incorporated into the $g_A^{eff}$. In case of the pnQRPA with nucleonic correlations, the $g_A^{eff}$ stands for the non-nucleonic correlations and nuclear medium effects, which may reduce the axial-vector NMEs.

One of the important non-nucleonic correlations are the isobar nucleon-hole contribution $\Delta N^{-1}$, which is a kind of the quark spin-isospin polarization of the nuclear medium. Experimentally one may test the non-nucleonic effect by comparing the summed axial-vector strength with the nucleon-based sum-rule limit as discussed in [7,8].

Actually, the DBD strength for the grand-state transition is only a small fraction of the order of $10^{-5}$ of the total DBD strength summed over all final states up around to 60 MeV, and thus accurate evaluations for them are interesting by taking into accounts exactly all possible effects such as non-nucleonic (meson, isobars), nuclear medium effects, momentum-dependent interactions, and others that affect the DBD NME. Here it is interesting to find that the recent calculation, which includes two-body currents, reproduces the large GT rates in simple double-magic nuclei [64].

The vector weak coupling $g_V$ may be also modified due to non-nucleonic correlations and nuclear medium effects, which are not explicitly included in the models. In fact, nuclear models used to evaluate vector-type DBD NMEs are not perfect, and thus NMEs $M_F^{0\nu}$ depend more or less on the models and the interaction parameters used in the models. So it is interesting to study experimentally vector-type forbidden single-$\beta$ NMEs and to compare them with theoretical NMEs to see if one needs some re-normalization coefficient $g_V^{eff}/g_V$ defined as the ratio of the experimental to theoretical vector-type NMEs [6]. Experimentally, extraction of the vector NMEs in non-unique forbidden $\beta$-decays, however, is hard. One way is to study analogous $\gamma$ transitions [55].

The nuclear re-normalization effect is a small fraction of the order of $10^{-6}$ of the total DBD strength, and thus not significant in the case of the super-allowed Fermi transition for the isobaric analog state with the large strength near the sum rule limit [65], but may not be negligible in the ground-state transitions associated with the isospin, spin, and nuclear-medium changes and the tiny strength of the order of $10^{-5}$ of the total strength.

Perspectives of experimental studies of single-$\beta$ NMEs and nuclear structures associated with the DBD NMEs are discussed elsewhere [66].

**Funding:** This research received no external funding.

**Acknowledgments:** The author thanks C. Agodi for valuable discussions and encouragements during the course of the present work, and the nuclear and leptonic CER colleagues for the collaboration works.

**Conflicts of Interest:** The author declares no conflict of interest.

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
