# Peer review of "Neutrino-Mass Sensitivity and Nuclear Matrix Element for Neutrinoless Double Beta Decay"

_universe, doi:10.3390/universe6120225_

Round 1
Reviewer 1 Report
This manuscript is devoted to discuss the nuclear matrix element (NME) for neutrinoless double beta decay, and the effect on neutrino mass sensitivity. The author augued that the experimental measurements of charge-exchange nuclear and lepton reactions would be useful to probe the NME and provide a model independent estimation of the NME. This results in this work is intersting and deserves the publication if the following issues can be clarifitied.
1) Although it might be fine to understand from the content, it is better to clarity the NME is for neutrinoless double beta decay, or the two-neutrino mode.
2) The momentum transfer for single beta decay, and two-neutrino double beta decay is around several MeV, but the momentum transfter for neutrinoless double beta decay can be as large as 100 MeV. With is big difference, how to make connection between these two kinds of reactions?
3) On the quenching of gA, there are some arguments that it is model dependent, and originated from limination of caculation power.
Could be please comment on the origin of gA quenching, and how to test it in experiments?
Author Response
The author thanks the referee for the valuable comments, which are used to improve the manuscript. The replies to the individual comments are as given in the file uploaded.

Reviewer 2 Report
Report of universe-1001158
--------------------------------------
The manuscript "Neutrino-mass sensitivity and nuclear matrix element for neutrinoless double beta decay" reviews the importance of a good knowledge of reliable nuclear matrix elements (NMEs) in order to anticipate the neutrino-mass sensitivity of future neutrinoless double-beta decay (0nbb) experiments, and discusses several experimental probes, such as two-neutrino double-beta decay, nuclear and leptonic charge-exchange reactions, and electromagnetic reactions that can be used to test the value of the NMEs. A section is devoted to the very important issue of the overestimation of NMEs in most nuclear many-body models (``quenching'').
The article is very interesting. It covers very carefully aspects such as the sensitivity of 0nbb experiments to the mass of the neutrinos in the case of normal and inverted neutrino mass hierarchies as the function of the NMEs, and how to probe these NMEs measuring other observables. I find the latter part especially illuminating, as the author connects 0nbb decay not only to other first and second order weak processes, but also to nuclear charge-exchange reactions and especially electromagnetic transitions, an aspect usually forgotten in the literature. Overall, the author shows clear command of all the subjects discussed.
The manuscript is well written, and even though I find the references too much centered on previous works of the author and some discussions should be extended, these are mostly small details. Once these aspects are taken care of (see below for more specific recommendations) I find that this work merits publication in Universe.
More specifically, there is a number of places where I find the manuscript should be improved by briefly mentioning additional works currently missing, or discussing with more care certain subtleties. My list of recommendations to improve the manuscript is as follows
1) The text claims
"The phase space can be well calculated"
but gives no references. Very precise calculations have been given by Kotila et al [PRC 85, 034316 (2012), PRC 87, 024313 (2013)] and Stoica et al. [PRC 88, 037303 (2013),Front. Phys. 7:12 (2019)].
2) Also, the text reads
"Recent works on the neutrino oscillations show that the neutrinos are massive particles"
again without any references. There is of course plenty of choices here, but citations to the pioneering works by SNO, SuperKamiokande or KamLAND seem mandatory.
3) Likewise, where the possible mechanisms triggering 0nbb are discussed
"[DBD] is due to several modes (α), the light neutrino-mass mode, the heavy neutrino-mass mode, the SYSY particle mode, the right-handed weak interaction mode, the Majoron-boson mode, and others"
no references are given. Again, review articles by Vergados, Simkovic, Faessler and many others have covered these modes in quite some detail, and it would be extremely useful for readers to find such references in the text.
4) In equation (14) I believe m_e (electron mass) is missing from the definition of Delta(i), see PRC 88, 037303 (2013). Also, I think that E_i should stand for the energy of the ith state with respect to the initial state of the transition, but at the moment this does not seem to agree with the manuscript, which refers to E_i simply as "the excitation energy".
5) Section 3.1 only discusses FSQP calculations of two-neutrino double-beta decay. These are of course very interesting, but I would find it more complete to at least mention that these decays have also been extensively explored with the QRPA (PRC 91, 054309 (2015), PRC 97, 034315 (2018), PRL 122, 192501 (2019)], shell model [PLB 711 (2012) 62, PRC 100 014316 (2019), PLB 797 (2019) 134885, PLB 802 (2020) 135192], IBM [PTEP 2013, 043D01] or effective field theory [PRC 98, 045501 (2018)].
6) The text correctly claims that
"GT NMEs with J π =1 + have been measured as shown in Fig.
7, and used to evaluate the 2-neutrino DBD NMEs"
However, an important point that should also be mentioned is that the phase between the two beta-decay legs cannot be determined in these kind of experiments, and therefore introduces an uncertainty in the evaluation.
7) In Sec. 3.3 values of gA^eff / gA ~ 0.5 are given. Since the ``quenching'' depends on the model used, can it be added to which model does the above value correspond?
8) In Sec. 4 the text claims that
"Accordingly, the quenching coefficients around gA^eff /gA ≈ 0.5
are universal for the wide momentum region of the DBD interest"
I am not sure all evidence points in the same direction. For instance, comparison to inelastic neutrino scattering on 12C does not seem to demand a large quenching [Prog. Part. Nuc. Phys. 59 (2007) 486], and a recent work on muon capture also finds good agreement to data for gA^eff ~ gA [PRC 102, 034301 (2020)]. In my opinion, the jury is still out, and the author should add some discussion on this subject to give a more balanced view.
9) The first paragraph in page 12 discusses the effect of gA^eff in QRPA 0nbb NMEs. However, an alternative sometimes used in the QRPA to reduce the value of beta and double-beta decay matrix elements without necessarily reducing the value of gA^eff is to reduce the value of the coupling g_pp (which corresponds to the isoscalar pairing interaction). Can the author briefly comment on this, specifically whether the behaviour seen in Fig.14 would remain if g_pp is varied?
10) In the summary, I do not fully agree with the statement
"Actually, there are no accurate theoretical-calculations on the g A value"
Even though all nuclear many-body calculations demand some approximations, I find very convincing the recent work Nat. Phys. 15 (2019) 428, where two kind of ab initio calculations reproduce well different Gamow-Teller transitions in medium-mass nuclei up to Sn100. At least in this case it seems to me that theoretical calculations describe well experimental data with gA^eff~gA. This work should be discussed in the manuscript, at least briefly.
11) I very much agree that studies on the vector weak coupling gV are very interesting. However, if gV^eff were different to gV, this presumably would have been observed in superallowed beta decays, which seem to be pretty well reproduced by the theoretical calculations of Towner and Hardy [eg J. Phys. G: Nucl. Part. Phys. 41 (2014) 114004]. Can a brief discussion be added to the text on this issue?
In summary, I think the manuscript covers very interesting topics and will be very useful to the community. Once the aspects above are improved, I will recommend the publication of the current work in Universe.
Author Response
The author is very grateful to the referee 2 for the very valuable comments. The replies to the individual comments are in the file up-loaded.
